# Single-Cell DNA Sequencing Reveals an Evolutionary Pattern of CHIP in Transplant Eligible Multiple Myeloma Patients

**DOI:** 10.3390/cells13080657

**Published:** 2024-04-09

**Authors:** Enrica Borsi, Ilaria Vigliotta, Andrea Poletti, Gaia Mazzocchetti, Vincenza Solli, Luca Zazzeroni, Marina Martello, Silvia Armuzzi, Barbara Taurisano, Ajsi Kanapari, Ignazia Pistis, Elena Zamagni, Lucia Pantani, Serena Rocchi, Katia Mancuso, Paola Tacchetti, Ilaria Rizzello, Simonetta Rizzi, Elisa Dan, Barbara Sinigaglia, Michele Cavo, Carolina Terragna

**Affiliations:** 1IRCCS Azienda Ospedaliero-Universitaria di Bologna, Istituto di Ematologia “Seràgnoli”, 40138 Bologna, Italy; 2Department of Medical and Surgical Sciences (DIMEC), University of Bologna, 40126 Bologna, Italy

**Keywords:** clonal hematopoiesis of indeterminate potential (CHIP), single-cell DNA sequencing, stem cells, multiple myeloma

## Abstract

Clonal hematopoiesis of indeterminate potential (CHIP) refers to the phenomenon where a hematopoietic stem cell acquires fitness-increasing mutation(s), resulting in its clonal expansion. CHIP is frequently observed in multiple myeloma (MM) patients, and it is associated with a worse outcome. High-throughput amplicon-based single-cell DNA sequencing was performed on circulating CD34+ cells collected from twelve MM patients before autologous stem cell transplantation (ASCT). Moreover, in four MM patients, longitudinal samples either before or post-ASCT were collected. Single-cell sequencing and data analysis were assessed using the MissionBio Tapestri^®^ platform, with a targeted panel of 20 leukemia-associated genes. We detected CHIP pathogenic mutations in 6/12 patients (50%) at the time of transplant. The most frequently mutated genes were *TET2*, *EZH2*, *KIT*, *DNMT3A*, and *ASXL1*. In two patients, we observed co-occurring mutations involving an epigenetic modifier (i.e., DNMT3A) and/or a gene involved in splicing machinery (i.e., SF3B1) and/or a tyrosine kinase receptor (i.e., KIT) in the same clone. Longitudinal analysis of paired samples revealed a positive selection of mutant high-fitness clones over time, regardless of their affinity with a major or minor sub-clone. Copy number analysis of the panel of all genes did not show any numerical alterations present in stem cell compartment. Moreover, we observed a tendency of CHIP-positive patients to achieve a suboptimal response to therapy compared to those without. A sub-clone dynamic of high-fitness mutations over time was confirmed.

## 1. Introduction

Clonal hematopoiesis (CH) is defined by the presence of an expanded clonal hematopoietic cell population due to an acquired mutation conferring a selective growth advantage [1,2,3,4,5,6]. Clonal hematopoiesis of indeterminate potential (CHIP) is a subset of CH defined by a clonal population of blood cells bearing a point mutation or short/deletion with a variant allele fraction (VAF) ≥ 2% in gene that is recurrently mutated in hematological malignancies like acute myeloid leukemia (AML), myelodysplastic syndrome (MDS), and myeloproliferative neoplasms, with *DNMT3A*, *TET2*, and *ASXL1* being the most frequently implicated genes [7,8,9,10]. All these genes, which include epigenetic modulators, have a broad array of cellular functions including signal transduction, pre-mRNA splicing, and DNA repair [11,12,13].

Most CHIP studies have focused on gene mutations including single-nucleotide substitutions and indels. However, CHIP, as defined by mosaic chromosomal alterations (mCAs) such as amplifications, deletions, and copy-neutral loss of heterozygosity (CNLOH), has also been reported, albeit at a much lower prevalence [14,15,16,17,18,19].

CHIP is associated with 0.5–1% per year risk of progression to non-plasma cell hematologic neoplasms, mainly MDS and AML [1,3,8]. CHIP is also associated with higher all-cause mortality, largely mediated by increased risk of cardiovascular disease, myocardial infarction, and stroke [7,20,21,22].

Moreover, CHIP is a marker of poorer prognosis in patients with non-Hodgkin lymphoma and multiple myeloma (MM) receiving autologous stem cell transplantation (ASCT) [10]. Mouhieddine et al. found that MM patients with CHIP had a higher progression rate post-ASCT compared to those without CHIP. Indeed, targeted sequencing of 629 patients prior to ASCT detected CHIP in 21.6% of patients, who showed an inferior overall survival (Hazard Ratio [HR] 1.34, *p* = 0.02) and progression-free survival (HR 1.45, *p* < 0.001) compared to the others. Notably, patients without long-term lenalidomide maintenance post-ASCT had worse outcomes compared to those receiving lenalidomide, regardless of CHIP status [23]. However, the precise impact of CHIP on MM outcome is still poorly understood. Indeed, in another study, the presence of CHIP linked with an age-related etiology, did not exhibit any significant prognostic relevance among MM patients in different clinical settings [24]. In one other study, Maia et al. demonstrated CHIP in 50% of MM patients with MDS-PA (MDS phenotypic alterations) versus one-fifth of MM patients lacking MDS-PA. This finding suggests that MDS-PA and CHIP are currently present at diagnosis rather than acquired after chemotherapy [25].

Treatment for MM has improved considerably over the last two decades leading to markedly enhanced clinical outcomes [26]. However, treatment with novel agents possibly combined with high-dose chemotherapy and/or radiotherapy for MM may induce CHIP or promote the selection of pre-existing CHIP clones in residual hematopoietic cells [27,28,29].

The aim of this study was to explore the prevalence of CHIP in mobilized CD34+ cells before subsequent ASCT in twelve newly diagnosed MM (NDMM) patients using single-cell analysis. We discovered that their clonal architecture was composed of a predominant clone, comprising 3% to 63% of the cells, harboring a single variant. Additionally, 1–2 minor sub-clones were identified, each comprising less than 20% of the cells. We also observed the presence of minor doublet-mutated sub-clones in two patients, affecting genes involved in epigenetic modulation (i.e., DNMT3A), splicing machinery (i.e., SF3B1), and tyrosine kinase receptor activity (i.e., KIT). Longitudinal analysis of paired samples revealed a positive selection of mutant high-fitness clones over time, regardless of their affinity with a major or minor sub-clone. Copy number analysis of the panel of all genes (i.e., 20 leukemia-associated genes) failed to show the presence of numerical alterations in the mutated cells analyzed. Notably, three CHIP+ patients relapsed after receiving lenalidomide maintenance therapy.

## 2. Materials and Methods

### 2.1. Patient Samples and Cell Preparation

All patients included in the study provided written informed consent for biological studies and were treated at a single center, according to either national guidelines (e.g., for those outside clinical studies) or treatment schedules provided by clinical trials. Nineteen samples derived from twelve newly diagnosed MM (NDMM) patients who underwent an autologous stem cell transplantation (ASCT) were obtained during standard diagnostic procedures. Hematopoietic stem cells (HSCs) (i.e., CD34+ cells), both from diagnostic bone marrow (BM) aspirates and apheresis products, were isolated with an immunomagnetic bead-based strategy (MACS system, Miltenyi Biotec, Auburn, CA, USA). The purity of positively selected CD34 cells was assessed by flow cytometry using a conventional antibody panel. BM CD138− cells were obtained by depletion of tumor cell contamination using magnetic-activated cell-sorting technology (MACS system, Miltenyi Biotec, Auburn, CA, USA) employing specific anti-CD138 microbeads. Selected cells were viably cryopreserved in Bambanker^®^ hRM cell-freezing medium (LYMPHOTEC Inc., Tokyo, Japan) and immediately stored for subsequent single-cell analysis.

### 2.2. Patients’ Clinical Outcomes

In this study were enrolled 12 NDMM patients who were treated with either bortezomib-based triplets incorporating an immunomodulatory drug (bortezomib–thalidomide–dexamethasone [VTD, *n* = 6] or VD and lenalidomide [VRD, *n* = 1] or daratumumab-based quadruplets (VRD plus daratumumab [DVRD, *n* = 2] and bortezomib–cyclophosphamide–dexamethasone (VCD) plus daratumumab [DVCD, *n* = 3]) as induction therapy before subsequent ASCT. All patients received maintenance therapy after ASCT (lenalidomide, *n* = 7; ixazomib, *n* = 2; daratumumab–lenalidomide, *n* = 2; and daratumumab–ixazomib, *n* = 1). Response to induction therapy, as evaluated according to the International Myeloma Working Group (IMWG) criteria, [30] was as follows: complete response (CR) or higher (2 patients), very good partial response (VGPR) (6 patients), and partial response (PR) (4 patients). Best response after ASCT was stringent CR (sCR) in 2 patients, CR and VGPR in 2 and 4 patients, respectively, and PR in the remaining 4 patients.

### 2.3. Single-Cell DNA Sequencing

Single-cell DNA sequencing was performed using the Tapestri^®^ platform (Mission Bio, Inc., San Francisco, CA, USA) and a catalog panel to assess recurrent CHIP. Overall, the designed panel included 20 genes and 127 amplicons of size ranging between 175 and 275 bp. Cryopreserved CD34+ cells from either BM or leukapheresis products, BM-CD138− cells, and BM cells post-ASCT were thawed and processed by the Tapestri^®^ platform, according to the manufacturer’s protocol. Briefly, cells were counted and diluted in Cell Buffer (Mission Bio, Inc.) to ~4000 cells/μL in a total volume of at least 50 μL. Approximately 100,000–120,000 cells were loaded onto the Tapestri microfluidic cartridge together with a lysis mix and encapsulation oil. Cells were emulsified and incubated at 50 °C prior to thermally inactivating the protease. The droplets containing encapsulated cell lysates were then microfluidically combined with targeted gene-specific primers, PCR reagents, and hydrogel beads carrying cell-identifying molecular barcodes using the Tapestri instrument and cartridge. The emulsions containing cell lysate barcoding beads and PCR reagents were then exposed to non-irradiating UV light for 8 min to cleave off barcode-containing forward primers from the barcoding beads prior to targeted PCR amplification. Following amplification, the emulsions were broken using perfluoro-1-octanol and the aqueous fraction was diluted in water and collected for DNA purification with Ampure XP reagent (Beckman Coulter, Inc., Brea, CA, USA, # A63880). Sample indexes and Illumina adaptor sequences were then added via a 10-cycle PCR, and the amplified material was then purified a second time with Ampure XP reagent. Purified sample libraries were analyzed on a DNA 1000 assay chip with a TapeStation 4200 (Agilent Technologies Inc., Santa Clara, CA, USA) and sequenced on an Illumina NexSeq550 (Illumina, Inc., San Diego, CA, USA) with 150 bp paired-end setting.

### 2.4. Pipeline Processing and Variant Filtering

Sequencing data were processed using Mission Bio’s Tapestri Pipeline. Data were analyzed using Mission Bio’s portal (https://portal.missionbio.com/) and Mission Bio’s Tapestri Insights software package (Version 2.2), and visualized using R software version 4.1. Data pre-processing for variant discovery was part of the Tapestri proprietary analysis pipeline. Low-quality cells and variants were removed when complying with any of these 6 filter parameters: (1) removal of genotypes in cells with quality < 30; (2) removal of genotypes in cell with read depth < 10; (3) removal of genotypes in cell with alternate allele frequency < 20; (4) removal of variants genotyped in <50% of cells; (5) removal of cells with <50% of genotypes present; and (6) removal of variants mutated in <1% of cells.

### 2.5. Variant Calling

To define final pathogenic somatic variants, we first removed polymorphous variants detected in all samples and then we selected those variants having the following criteria: (1) mutated (if the variant allele frequency > 0.01) in >1% of cells; and (2) the minimum variant allele fraction (VAF) among the mutated cells is at least 0.2. These selected variants were manually reviewed and only exonic variants were kept. We used the DANN score system as the first filter step to assess the pathogenicity of a given variant (i.e., we used a cut-off DANN score of ≤0.8 to filter out all the non-pathogenic variants). From the good-quality variants, we selected non-intronic and somatic variants for downstream analysis. Briefly, pathogenic variants were labeled as such if: (1) they caused a biological loss of function (e.g., missense, nonsense, and frameshift), or (2) were reported as pathogenic in clinical databases (e.g., CLINVAR, COSMIC, and Varsome [31]) and had strong evidence of pathogenicity from in silico prediction tools. After filtering, we focused only on the pathogenic detected variants by excluding common variants in human population (freq. > 1%) and retaining only variants with confirmed evidence of pathogenicity in comprehensive clinical databases (i.e., COSMIC and CLINVAR) and in silico prediction tools (i.e., MutationTaster, MutationAsssesor, Polyphen2, SIFT, Provean, LRT, FATHMM, MetaSVM, and MetaLR), as annotated by ANNOVAR—dbNSFP (dbnsfp33a).

### 2.6. Clonal Architecture Analysis

Clonal architectures were initially determined by genotype clustering analysis including zygosity information with the Tapestri Insight software package and visualized using custom pipelines in R software. We included somatic coding (nonsynonymous) variants and cells with complete genotypes, whereas sub-clones with a missing genotype and with <1% of cells were removed. The minimum clone size was 10 cells, and variants only present in such small clones (<10 cells) were therefore removed. The variant metrics (VAF, genotype quality, and read depth) of each clone were inspected to identify small clones that were likely the result of allele dropout (ADO). Small clones below the ADO sample rate (cut-off < 1%) genotyped as wild type (WT) or homozygous for a given variant, but with decreased quality and read depth, were considered false positives, and the cells were appointed to the closest clone.

### 2.7. Copy Number Analysis (CNAs)

The copy number analysis was mainly based on Mission Bio’s Mosaic package version 2.3. The per-amplicon read counts were normalized to correct for systemic artefacts, first within the same cell across different amplicons by mean read depth, and then within the same amplicon across different cells by median read depth. This step was performed using the normalize_reads method from the Mosaic package. Normalized reads were converted to copy number estimates by assuming that the WT clone (as assessed by mutation analysis) was diploid and all the other cells normalized to read count were scaled to the counts of this clone to derive copy number estimates. This step was performed using the compute_ploidy method from the Mosaic package. Note the median read depth across different cells only considered good-quality cells, which are defined as those with at least 1/10 of the number of reads as that of the cell with the 10th rank in terms of read count.

Cells were clustered according to the grouping of the variants and per-gene CNAs were called for every clone only if *n* > 2 amplicons of a given gene showed an amplificated (CN > 2.5) or deleted (CN < 1.5) copy number signal.

## 3. Results

### 3.1. Single-Cell DNA Sequencing Panel Performance

To define the mutational profile of CHIP in MM patients, we performed a high-throughput amplicon-based single-cell DNA sequencing using the Tapestri^®^ platform. Our approach utilized a 127-amplicon panel (size range between 175 and 275 bp) covering 20 of the most frequently mutated genes associated with CHIP. The genes panel included splicing factor genes (i.e., *SF3B1*, *SRSF2*, *U2AF1*), genes implicated in epigenetic regulation (i.e., *TET2*, *IDH1/2*, *ASXL1*, *EZH2*, and *DNMT3A*), and known oncogenes/genes involved in cell signaling/transcription regulation (i.e., *TP53*, *FLT3*, *NRAS*, *KRAS*, *RUNX1*, *C-KIT*, *JAK2*, *GATA2*, *PTPN11*, *WT1*, and *NPM1*) (Appendix A). Tapestri^®^ provides a two-step droplet microfluidics approach that enables the identification of genomic alterations (single-nucleotide variants [SNVs] and small indels) across thousands of cells at single-cell resolution and in a targeted amplicon-based manner. Overall, we sequenced a total of 105,743 cells obtained from 19 samples collected at different time points from 12 ASCT-eligible patients with NDMM who underwent peripheral blood stem cell (PBSC) mobilization. In all of them, CD34+ cells were isolated from leukapheresis products. In addition, CD34+ bone marrow (BM) cells were collected at diagnosis in four of the twelve patients, while in three additional patients, sequencing analyses were performed either on mobilized CD34− cells (i.e., CHIP#1) or BM mononuclear cells (MNCs) collected during lenalidomide maintenance therapy (i.e., CHIP#11) or BM-CD138 negative cells (i.e., BM depleted by tumor cells) collected at relapse (i.e., CHIP#1) (as summarized in Figure 1A). We obtained a median throughput of 4972 cells per sample (interquartile range [IQR]: 2759–8385) (Appendix A), and a median sequencing coverage of 44 reads per amplicon per cell (IQR: 36X-122X) (Appendix A). Moreover, sequencing reads were evenly distributed across the 127 amplicons in a gene-independent manner. Importantly, all panel amplicons in each sample had sufficient coverage for variant annotation (Appendix A). The estimated median allele dropout (ADO) rate was 9.1% (IQR: 7.5%-15%; Appendix A).

From the variants annotated for each sample, we initially filtered out low-quality (median per sample 139, IQR: 95–551) and low-frequency (median per sample 150, IQR: 90–3779) variants using the Tapestri Insight pipeline filters (as detailed in the Section 2). Following this filtering step, we identified a total of 235 coding variants (including variant annotated as polymorphic) across the 12 patients, with a median of ten variants per patient (IQR: 8–14), and a median of eight variants per gene (IQR: 3–25) (Figure 1B and Appendix A). *TET2* was the most frequently mutated gene (64/235 SNVs, 27%), followed by *KIT* (27/235 SNVs, 11.5%), *EZH2* (27/235 SNVs, 11.5%), *DNMT3A* (25/235 SNVs, 10.6%), and *ASXL1* (22/235 SNVs, 9.4%) as shown in Figure 2A.

Among all the 235 variants identified, we found a total of 109 unique coding mutations in 18 CHIP-related genes across 12 MM patients (Figure 2B). Among those variants, 100 (91.74%) were SNVs, including 71 missense, 26 synonymous, and 3 nonsense variants; 6 (5.51%) were small indels with either frameshift (*n* = 5) or in-frame (*n* = 1), and 1 (0.92%) was associated with splicing (Figure 2B). Moreover, SNVs present in the 3′-untraslated regions of the *NPM1* gene (*n* = 2, 1.83%) were also observed. Specifically, most of the missense mutations were detected in the *TET2* gene (*n* = 17, 24%), followed by *ASXL1* (*n* = 11, 15%), *DNMT3A* (*n* = 10, 14%), and *TP53* (*n* = 7, 10%) (Figure 2C).

### 3.2. Single-Cell Mutational Spectrum of Genes Associated with Clonal Hematopoiesis in Harvested CD34+ Cells following PBSC Mobilization Therapy

To define final pathogenic somatic variants, we firstly removed polymorphous variants detected in all samples as described in the Section 2. We used a cut-off DANN score of ≤0.8 to filter out all the non-pathogenic variants, and only somatic variants were kept for downstream analysis. For those variants that did not report as pathogenic in clinical databases (e.g., CLINVAR and COSMIC), but with strong evidence of pathogenicity with a DANN score ≥ 0.8, in silico prediction tools were used to assess pathogenicity. Overall, most of the candidate variants that passed pre-pathogenic filtering criteria were further excluded, yielding to 0 to 3 pathogenic variants per patient (*n* = 11 pathogenic variants). We identified 6 patients out of 12 (50%) harboring detrimental somatic mutations in CHIP-related genes in CD34+ cells selected from leukapheresis products.

A single pathogenic CHIP mutation was detected in 2 of the 6 patients (33%), while 4 (67%) had 2 or more mutations in different genes (Figure 3A). Consistent with prior studies [4,10] 40% of DNMT3A mutations were truncating and 60% were missense (Figure 3B). The KIT, SF3B1, TET2, GATA2, and EZH2 mutations were all missense. For these patients (hereinafter called CHIP+), cells were grouped into clones based on the genotype call of the selected variants. The number of clones was carefully reviewed and crossed with zygosity and quality metrics information (i.e., VAF, genotype quality, and read depth) to identify false sub-clones resulting from ADO. Because this study was based on de novo variant discovery, we used a cut-off ADO rate of 1% for low-frequency clone calling (all clones below this threshold were filtered out). Specifically, for 2 patients we could detect 1 clone (33%), whereas 4 patients had 2 or 3 clones (67%), with *DNMT3A* being the most frequently mutated gene across all patients (5/6, 83%) (Figure 3A,C).

Moreover, we observed a patient-specific mutational profile at the time of ASCT with varying mutational load, ranging from only one variant in the CHIP#7 patient to three variants in the CHIP#1 patient. In two CHIP+ patients, we detected at least two somatic and nonsynonymous variants that were present in 63% and 78% of the genotyped cells (CHIP#9 and CHIP#11, respectively), defining the major clone (Appendix A). Interestingly, we identified sub-clonal variants at intermediate frequencies in CHIP#12 and CHIP#9 (mutated in 15.6% and 19% of the cells, respectively) and a substantial number of minor sub-clonal variants (mutated in 1–7% of the cells) as shown in Figure 3D. Remarkably, 4 of the 6 CHIP+ cases showed various mutations at very low frequency, indicating the common presence of minor sub-clones in the CD34+ stem cell compartment at the time of ASCT, and the capability of this panel for single-cell DNA sequencing to detect mutations in such small populations.

Additionally, 5 out of 6 CHIP+ patients harbored at least one *DNMT3A* variant in expected hotspots (i.e., DNMT3A p.S881R and DNMT3A p.R882H in CHIP#9 and CHIP#11, respectively). Variants in the *DNMT3A* gene (coding for the SAM-dependent MTase domain) were mostly single amino acid substitutions (i.e., DNMT3A p.R749C in the CHIP#14 patient), except for one nonsense mutation resulting in the premature stop codon (DNMT3A p.R771* in patient CHIP#7). Moreover, we identified one patient (CHIP#1) harboring a splicing variant detected in an authentic splice site leading to exon skipping, and likely in a protein loss of function (DNMT3A c.2323-2A>G). Additionally, a serine-to-tyrosine substitution (i.e., KIT p.S821Y) in both CHIP#9 (characterizing the major clone) and CHIP#12 (representing the minor clone) patients were detected (Figure 4 and Appendix A).

Single-cell technology clearly revealed the co-occurrence and mutual exclusivity of driver mutations at the cellular level. Therefore, our subsequent investigation focused on all CHIP mutations at clonal resolution, revealing that the clonal architecture of MM patients’ CD34+ cells in the apheresis product typically consists of a predominant clone harboring just one variant encompassing 3% to 63% of the cells, along with 1 to 2 minor sub-clones, each representing less than 20% of the cells (Figure 4).

Notably, in two patients we observed co-occurring mutations (i.e., cellular-level co-occurrence) involving an epigenetic modifier (i.e., *DNMT3A*) and/or a gene involved in splicing machinery (i.e., *SF3B1*) and/or tyrosine kinase receptor (*KIT*) in the same clone. Indeed, CHIP#9 is carrying a minor doublet-mutant clone DNMT3A p.S881R/KIT p.S821Y accounting for 5% of all genotyped cells, whereas CHIP#11 had a small sub-clonal doublets-mutant representing 1% of all genotyped cells (DNMT3A p.R882H/SF3B1 p.K700E) (Figure 4). Among the list of variants, all the pathogenic SNVs identified were heterozygous mutations.

Collectively, these data demonstrate that CHIP is common in apheresis products from patients with NDMM.

### 3.3. Analysis of CHIP Clonal Dynamics in Longitudinal Samples Reveals a Positive Selection of Mutated Cells during Disease Progression

To investigate the clonal evolution of CHIP clones at different time points, we collected and analyzed longitudinal samples for four MM patients. The presence of CHIP mutations was determined in various specimens obtained either before starting induction therapy (i.e., CD34+ cells from diagnostic BM aspirate, *n* = 3 [for patient CHIP#11, CHIP#12 and CHIP#14]); or after ASCT (i.e., BM MNCs during maintenance phase, *n* = 1 [CHIP#11], and BM-CD138- cells at relapse phase, *n* = 1 [CHIP#1]).

Overall, our findings suggest that in the majority of cases, the driver mutation (i.e., major clone) detected before PBSC mobilization therapy at the time of disease onset had expanded and/or persisted at the time of transplant and/or relapse phase.

Specifically, for CHIP#1, a total of 12,535 cells were sequenced across two time points (3625 and 8910 for each time point, respectively), with good panel uniformity and an overall coverage of 35×–45× per cell per amplicon. The CHIP#1 patient exhibited three independent clones in the stem cell compartment at the time of ASCT. One clone carried DNMT3A c.2323-2A>G alone (comprising the major clone, C1 clone), which was significantly enriched in the BM at relapse phase (Fisher’s exact test *p* < 0.00001 for T1 vs. T2). Furthermore, there was evidence of two additional minor sub-clones: one carried GATA2 p.A372V (C2 clone) alone, and a third carried the EZH2 p.Q533P (C3 clone) mutation (Figure 5A,B and summarized in Appendix A). Both sub-clones underwent significant expansion in the BM-CD138− fraction at the time of relapse (Fisher’s exact test, +0.98% *p* = 0.00194 and +3.50% *p* < 0.00001 for C2 and C3, respectively). The GATA2 mutant clone was not observed in the CD34 negative cell fraction collected following PBSC mobilization therapy but was still present in a small cell population at the time of relapse (BM CD138 negative fraction) (Figure 5C,D and summarized in Appendix A).

For CHIP#11, a total of 18,356 cells were sequenced across three time points (1028, 10,660, and 6668 for each time point, respectively) with good panel uniformity and an overall coverage of 32x–57x per cell per amplicon. This patient had a more complex genetic architecture, with evidence of three different clones with the major clone C1 (i.e., DNMT3A p.R882H) comprising almost 38% of the CD34+ in the BM at the time of disease onset (BM CD34+ cells, diagnostic sample). Notably, the major clone was significantly enriched in all paired samples analyzed (41.44% in CD34+ cells at the time of ASCT and 50.42% in BM cells post-ASCT during maintenance therapy, respectively) (Fisher’s exact test *p* = 0.0154 for T0 vs. T1 and *p* < 0.00001 for T1 vs. T2, respectively, Figure 6A), whereas the C3 clone harboring the NRAS p.Q61R variant present in BM CD34+ cells was no longer detectable at disease progression (Fisher’s exact test *p* < 0.00001, and *p* = 1.00 for T0 vs. T1 and T1 vs. T2, respectively, Figure 6A). The C2 minor sub-clone (DNMT3A^Het^/SF3B1^Het^ double mutant clone) diminished in the CD34+ at the time of PBSC mobilization (Fisher’s exact test *p* = 0.00014 for T0 vs. T1) but persisted as minor clones throughout the remaining time points (Fisher’s exact test *p* < 0.00001 for T1 vs. T2) (Figure 6A and summarized in Appendix A).

For CHIP#12, a total of 9539 cells were sequenced across two time points (4029 and 5510 for each time point, respectively) with good panel uniformity and an overall coverage of 38×–78× per cell per amplicon. This patient had evidence of a minor KIT^Het^ single mutated CHIP clone at the time of disease onset that was still present and significantly enriched in the stem cell apheresis product (7.4% and 15.6% of all genotyped cells, respectively) (Fisher’s exact test *p* < 0.00001 for T0 vs. T1, Figure 6B).

Lastly, for CHIP#14, a total of 14,616 cells were sequenced across two time points (8385 and 6231 for each time point, respectively) with good panel uniformity and an overall coverage of 33×–44× per cell per amplicon. Among the list of variants, two pathogenic mutations were identified with high genotype quality. Indeed, this patient had two independent clones affecting epigenetic modifiers (i.e., TET2^Het^ and DNMT3A^Het^) detected in both the diagnostic and apheresis samples with significant fluctuation of these clones over time (Fisher’s exact test *p* = 0.00444 for C1 and *p* < 0.00001 for C2, respectively Figure 6C and summarized in Appendix A).

Overall, these data exemplify distinct clonal architectures at diagnosis and during disease progression, highlighting the feasibility of the technology to detect small sub-clones.

### 3.4. Copy Number Alterations (CNAs) in CHIP+ Samples

CHIP is currently defined as a clonal hematopoietic population carrying somatic point mutations in one of the leukemia-associated genes. Patients with MM often present with chromosomal abnormalities in addition to somatic point mutations. To determine if chromosomal abnormalities can co-occur with point mutations as part of CHIP in our cohort, in addition to studying the dynamics of clonal SNVs, the copy number alterations (CNAs) of all 20 genes in the panel (127 amplicons) were studied.

Normalized counts of sequencing reads using the amplicons in the panel were used to calculate CNAs for each amplicon locus tested. Using the wild-type (WT), heterozygous (HET), and homozygous (HOM) genotypes called by Tapestri Pipeline software version 2.2, clones with no mutations (WT) were assumed to also present a normal diploid copy number and were thus used as a reference to compute potential CNAs present in mutated clones. A possible limit of the approach is that WT clones are not guaranteed to be diploid for CNA, since WT definition was obtained by mutation analysis. Nonetheless, this normal reference choice bias is forced by the fact that a true diploid cell population was not available in this study. As a consequence of this reference choice, it is important to note that the CNA analysis has an intrinsic inability to detect possible CNAs affecting the majority of cells in given samples.

Overall, we did not observe any loss or gain of gene copies in any CHIP+ samples analyzed involving the mutated loci (Figure 7A). These data suggest that CHIP, at least in our cohort, is solely defined by the presence of somatic point mutations and not by a co-occurrence of CNAs events in mutated cells involving just the mutated loci.

### 3.5. Clonal Architectures and Clinical Outcomes of CHIP+ Patients

The median age of all patients in our cohort was 62 years (range: 45–70) at diagnosis and 63 years (range: 46–70) at the time of ASCT (Table 1; for cytogenetic characteristics and numerical alterations please refer to Appendix A for detailed information, respectively). In 4 of the 6 CHIP+ patients, induction therapy comprised a three-drug regimen incorporating bortezomib combined with an immunomodulatory agent, while the remaining 2 patients received a quadruplet including an anti-CD38 monoclonal antibody. In the group of CHIP− patients, those receiving triplets and quadruplets numbered three each. All patients in the CHIP+ group were offered single-agent maintenance therapy with lenalidomide (*n* = 4) or ixazomib (*n* = 2) following ASCT, while in 3 patients in the CHIP− group, maintenance therapy included a daratumumab-based doublet (Table 1).

Response to therapy was noted for simple numerical purposes, and is summarized in Figure 7B. Basically, all the 12 patients in both CHIP+ and CHIP− groups achieved at least a PR after induction therapy, including VGPR or higher in 4 patients in each of the two groups. After ASCT, one CHIP+ patient and three CHIP− patients achieved CR or higher (Figure 7B). Three CHIP+ patients relapsed during maintenance therapy with lenalidomide, and clonal evolution could be investigated in two of them. Specifically, the CHIP#1 patient achieved a VGPR following VTD induction therapy and subsequent ASCT. After 11 months from starting maintenance therapy with lenalidomide (and 21 months from ASCT), he relapsed. In this case, the major C1 clone detectable at ASCT was significantly enriched at relapse (i.e., DNMT3A c.2323-2A>G, with an increase of +8.54%, *p* < 0.00001 by Fisher’s exact test). Although C2 and C3 clones were classified as minor sub-clones (GATA2^Het^ and EZH2^Het^, respectively), they appeared to be significantly expanded in all time points analyzed, with an increase of +0.98% and +3.50% for C2 and C3, respectively (Fisher’s exact test *p* = 0.00194 and *p* < 0.00001 for C2 and C3, respectively).

CHIP#11 was a 69-year-old woman at disease onset, who received a VTD induction therapy (achieving a VGPR response), and lenalidomide as a maintenance regimen. After achieving a VGPR response post-induction, she relapsed after 19 months during lenalidomide maintenance. She was already harboring a major C1 clone at disease onset; this was not erased by the therapy but was significantly enriched over time, being present in the BM cells 7 months after starting maintenance therapy (i.e., DNMT3A p.R882H). Indeed, we found significant differences detected by Fisher’s exact test for the C1 clone, with an increase of +3.90% and +8.90% over time (*p* = 0.0154 for T0 vs. T1 and *p* < 0.00001 for T1 vs. T2, respectively).

These data, although preliminary, might suggest that high-fitness CHIP mutant clone(s) at the time of ASCT, or even before, at the time of diagnosis, are more likely to be positively selected over time.

## 4. Discussion

Hematopoietic stem cells (HSCs) naturally accumulate various somatic mutations over time, some of which confer selective advantages in a context-dependent manner [3,32]. The term clonal hematopoiesis of indeterminate potential (CHIP) defines a phenotype where hematopoietic cells carrying somatic leukemic-driver mutations expand clonally without apparent hematological disease [7]. CHIP is prevalent in plasma cell neoplasms, with up to 30% occurrence in treated MM patients as the incidence of MM rises with age [33]. This association between CHIP and MM may result from age-related changes in HSCs, with some patients exhibiting CHIP even without MM [33].

To the best of our knowledge, this is the first study employing a single-cell-based approach to detect, in the stem cell compartment, mutations commonly linked to CHIP in NDMM patients. We used the Tapestri technology to analyze the CHIP mutational profile at the time of ASCT and to explore clonal evolution from the disease onset throughout subsequent treatment phases, including maintenance. Of the twelve MM patients analyzed, six harbored CHIP pathogenic mutations in CD34+ cells collected after PBSC mobilization. This incidence exceeds the rates commonly reported in the literature, which is possibly attributable to the heightened sensitivity of the employed assay. Consistent with earlier reports, *DNMT3A* was the most frequently mutated gene, with mutations predominantly located in the methyltransferase catalytic domain, potentially impacting its methylation activity and leading to loss of function [34]. Additionally, *DNMT3A* mutations have been linked to a higher risk of developing hematological disease and to poorer outcomes in both AML and MM patients [33,35,36,37,38].

Overall, two possible scenarios of CHIP possibly influencing MM progression have been described. In the first, mutant HSCs with CHIP may give rise to B cells that subsequently acquire additional mutations, leading to a plasma cell malignancy. Once the malignant state has been reached, the presence of CHIP may then alter and influence the disease phenotype, increase the risk of therapy-related malignancies (tMNs) and all-cause mortality, and, ultimately, influence response to therapy [39]. In support of this scenario, Diamond et al. recently postulated two modes for tMN evolution, both involving the selection of pre-existing CHIP clones by chemotherapy [28]. In the second scenario, the presence of CHIP in mutant cells may modify the secretion of inflammatory cytokines and, therefore, the local BM niche, thus contributing to the progression of MM [39,40]. Moreover, it has been postulated that CHIP might influence cancer progression and recurrence, possibly due to cell–cell interactions between CHIP clones and cancer cells [41], impacting the immune function on immune surveillance and promoting an increased inflammatory milieu due to the clonal expansion of effector cells, or reduced tolerance toward cancer-directed therapy [42,43]. Indeed, myeloid cells carrying mutations in *DNMT3A* and *TET2* can stimulate inflammation through upregulation of IL-1β and IL-6 [21,44].

Consistent with this observation, we analyzed three MM patients who experienced relapse during maintenance therapy, all of them having a *DNMT3A* mutant clone. It is possible that the induction agents used in the treatment of MM might have contributed to creating a proinflammatory microenvironment, leading to a positive selection of CHIP clones. Indeed, the CHIP#11 patient already had evidence of a *DNMT3A* mutant clone at disease onset on BM-CD34+ cells, and we observed a significant increment of the CHIP clonal burden over time; on the contrary, in the CHIP#1 patient, the major clone persisted in the BM even at the time of disease progression. We might speculate that a proinflammatory microenvironment may have impaired the HSCs’ fitness, leading the CHIP clones to have a selective re-population advantage, particularly under exogenous stress condition such as transplantation; alternatively, as recently suggested, the pre-existing CHIP clone(s) might have escaped to the chemotherapy exposure via apheresis and reintroduced by ASCT [28]. Here, we present corroborative evidence through the identification of CHIP clones at the onset of the disease, coupled with the observation of an increased prevalence of these clones in the apheresis products. We acknowledge that with the data produced we cannot rule out both possibilities, and this will be the subject of future analyses.

Notably, we found evidence of significant changes, either positive or negative, in CHIP clones’ size across disease progression. Among mutations detected along the analyzed time points (i.e., BM CD34+ at the diagnosis, CD34+ collected following PBSC mobilization therapy, and BM cells during maintenance or at relapse), the majority had increased VAF (41%, 5/12), whereas 25% remained stable over time (detected in 3 clones out of 12), 17% (2/12) had decreased in VAF, and, finally, the remaining emerged at disease progression (17%, 2/12).

These results strongly suggest that pre-existing CHIP mutants are advantaged to expand and persist under selective pressure.

It has been demonstrated that single-cell multi-omics of DNMT3A p.R882 CD34+ HSC populations in MM patients revealed a myeloid-biased differentiation and megakaryocytic expansion within humans, suggesting that broad transcriptional consequences of *DNMT3A* mutation affect differentiation, even in the absence of a strong exogenous source of inflammation [45]. Therefore, we cannot rule out the possibility that some mutant clones carrying a high-fitness mutation might expand in different settings regardless of inflammation status and/or external stimuli. Future studies employing a large cohort of patients with sequential sampling before, during, and after therapy will be needed to deeply characterize the kinetics of CHIP.

CHIP has been associated with all types of hematological malignancy, including MM. It is well recognized that the presence of more than one mutation and increased mutational burden both positively predict for malignant progression, as well as for a general increased risk of death (not necessarily cancer related). Therefore, rapid advances in the biologic and clinical understanding of CHIP may reveal a critical opportunity to improve patient care, especially in the era of new cancer research, focusing on the development of immunotherapies to prime, augment, or sustain immune responses.

To date, the impact of immunotherapies on CHIP remains controversial, with some studies indicating that immunotherapies may promote expansion of CHIP clones but others reporting no impact on clonal expansion [46,47]. In the MM setting, Miller et al. have demonstrated that CHIP has the potential to influence chimeric antigen receptor (CAR) T cell biology and activity through multiple mechanisms [47].

## 5. Conclusions

In summary, by employing single-cell technology, we successfully identified rare pathological variants within the compartment of CD34+ cells collected and subsequently re-infused in NDMM patients. A sub-clone dynamic of high-fitness mutations over time was confirmed. These significant findings suggest that prospectively monitoring the dynamics of CHIP mutations could greatly benefit the clinical management of MM patients. The detection of CHIP in routine clinical practice could have far-reaching implications, including enabling more accurate risk assessment and the potential for early interventions that could modify disease risk. If therapeutic suppression of these clones becomes achievable, resulting in a reduced cancer risk and an improved overall survival, the role for broad screening for CHIP may become unequivocal.

## Figures and Tables

**Figure 1 cells-13-00657-f001:**
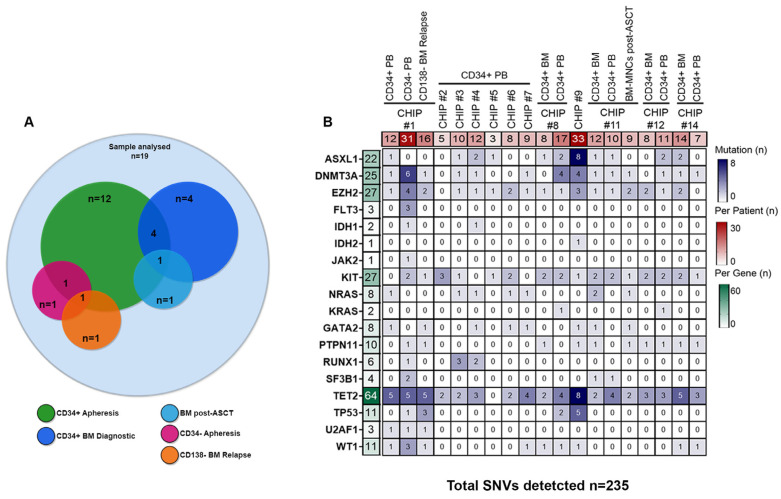
Single-nucleotide variants (SNVs) detected by single-cell DNA sequencing of 12 transplant-eligible patients with NDMM who underwent PBSC mobilization. (**A**) Venn diagram of the study design. In total we analyzed 19 samples collected at the following time points: BM CD34+ cells at diagnosis (*n* = 4), circulating CD34+ (*n* = 12) or CD34− (*n* = 1) cells selected from leukapheresis products before ASCT, BM-MNCs during post-ASCT maintenance therapy with lenalidomide (*n* = 1), and BM-CD138− cells at the time of relapse (*n* = 1). (**B**) Heatmap with the oncoprint of patient samples analyzed by single-cell DNA sequencing. Each column represents a patient, and each row represents a gene with at least 1 detected coding variant (235 different variants across 18 genes). The total number of variants detected per gene is shown in the left column with a green color gradient and the number of variants detected per patient is shown in the upper row with a red color gradient. The heatmap blue color gradient indicates the number of variants per patient and gene. CD34+ BM = CD34+ cells from bone marrow at diagnosis; CD34+ PB = CD34+ cells from apheresis product; CD34− PB = CD34− cells from apheresis product; BM-MNCs post-ASCT = bone marrow mononuclear cells post-autologous stem cell transplantation; CD138− BM relapse = CD138− cells from bone marrow at relapse.

**Figure 2 cells-13-00657-f002:**
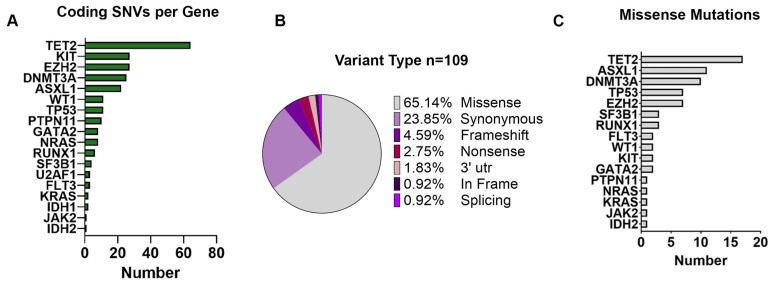
Type of unique single-nucleotide variants detected in MM patients. (**A**) Coding SNVs organized by gene. Histograms show the number of variants identified per gene in all the 19 samples analyzed. (**B**) Pie chart with the distribution of the 109 unique variants according to the protein translation implication (missense, synonymous, frameshift, nonsense, 3′ utr, in frame, or splicing). (**C**) Histograms show the number of missense mutations identified per gene in all of the 19 samples analyzed. All variants reported are mutated in at least 0.5% of the cells from a given sample.

**Figure 3 cells-13-00657-f003:**
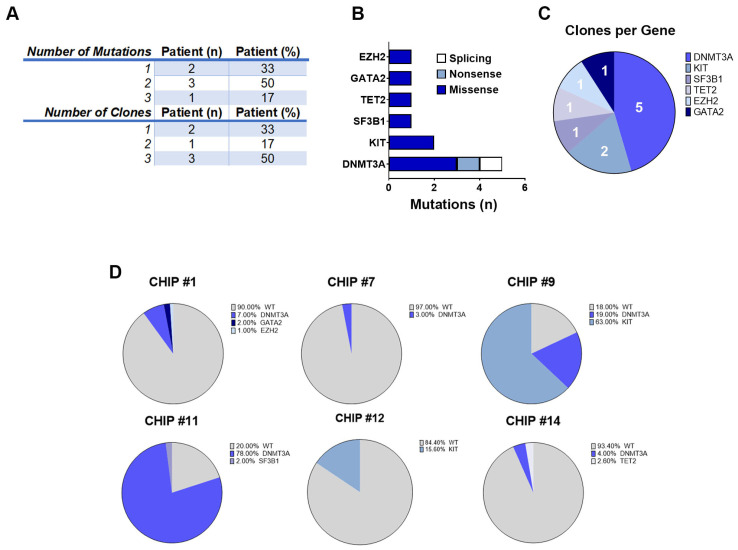
CHIP clones identified in CD34+ cells selected from leukapheresis products in 12 NDMM patients. (**A**) Table shows the number of mutations and clones identified in CD34+ cells collected following PBSC mobilization therapy in MM CHIP+ patients. (**B**) Histograms show the number of mutations in CHIP+ patients organized by gene. (**C**) Pie chart with the distribution of the clones organized by gene. (**D**) Pie charts with the distribution of clones for all the CHIP+ patients.

**Figure 4 cells-13-00657-f004:**
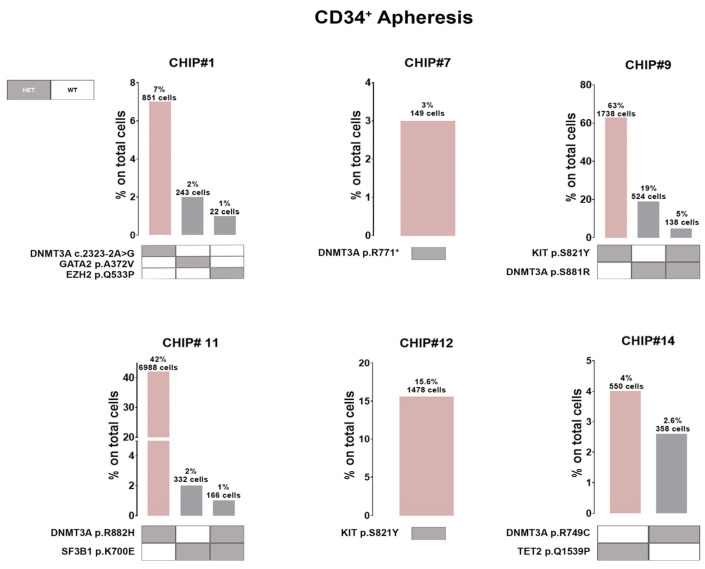
Clonal architecture of CHIP clones in MM patients at the time of transplant. Bar plots show the co-occurrence of mutations in CD34+ cells collected by leukapheresis in the six CHIP+ patients. Grey boxes represent the genes with heterozygous (HET) mutations whereas wild-type (WT) genotypes are shown with white boxes.

**Figure 5 cells-13-00657-f005:**
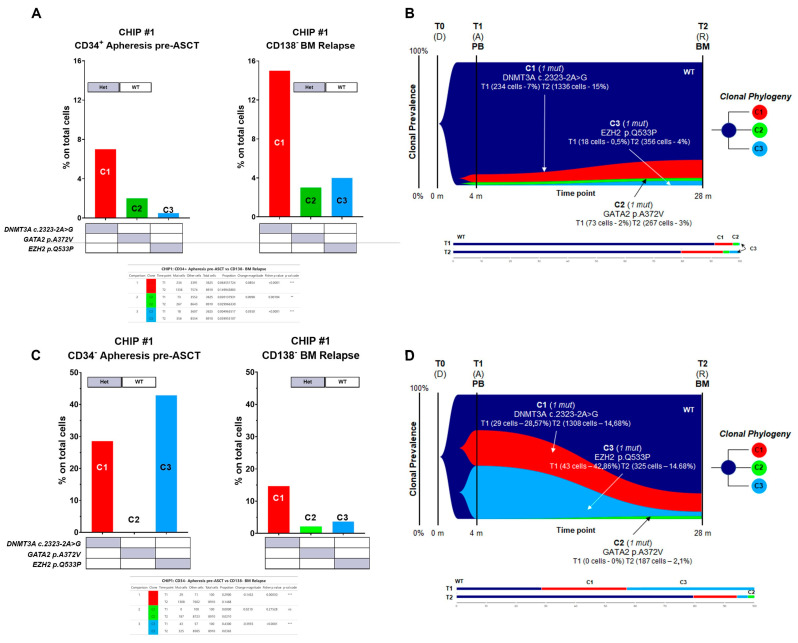
Genetic architecture of CHIP#1 patient. Bar plot showing co-occurrence of mutations in (**A**) CD34+ apheresis-derived cells and in CD138− at relapse phase and (**B**) CD34× apheresis-derived cells and in CD138× at relapse phase for CHIP#1 patient. Grey box represents the genes with heterozygous (HET) mutations whereas wild-type (WT) genotypes are shown with a white box. In panels (**C**,**D**), fish plot showing acquisition of mutations in serial time points. Bar plots show the distribution of cells in the sub-clones. At relapse phase, the major clone was significantly enriched. Minor clones include cells with a different heterozygous combination of mutations of *DNMT3A*, *GATA2*, and *EZH2*. Significant differences detected by Fisher’s exact test are shown in the tables below the histograms. T0 (D) = Diagnosis, T1 (A) = Apheresis, T2 (R) = Relapse, BM = Bone marrow, PB = Peripheral blood. ** < 0.001, *** < 0.001 and ns = non-significant.

**Figure 6 cells-13-00657-f006:**
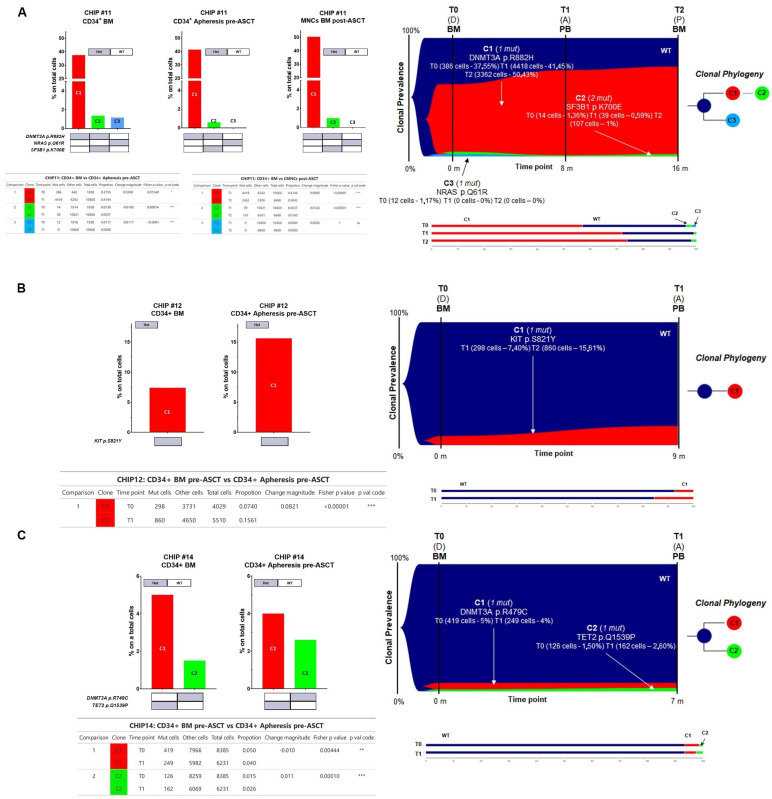
Genetic architecture of CHIP#11, CHIP#12, and CHIP#14 patients. In panel (**A**–**C**), a bar plot showcases the co-occurrence of mutations at different time points analyzed, including CD34+ BM cells at disease onset (CHIP#11, CHIP#12, and CHIP#14, respectively), CD34+ apheresis-derived cells, and BM cells post-transplant (CHIP#11) (left panel). Grey box represents the genes with heterozygous (HET) mutations whereas wild-type (WT) genotypes are shown with a white box. Fish plot showing acquisition of mutations in serial time points (right panel). Bar plots show the distribution of cells in the sub-clones. Significant differences detected by Fisher’s exact test are shown in the tables below the histograms. T0 (D) = Diagnosis, T1 (A) = Apheresis, T2 (P) = BM post-ASCT, BM = Bone marrow, PB = Peripheral blood. * < 0.05, ** < 0.001, *** < 0.001 and ns = non-significant.

**Figure 7 cells-13-00657-f007:**
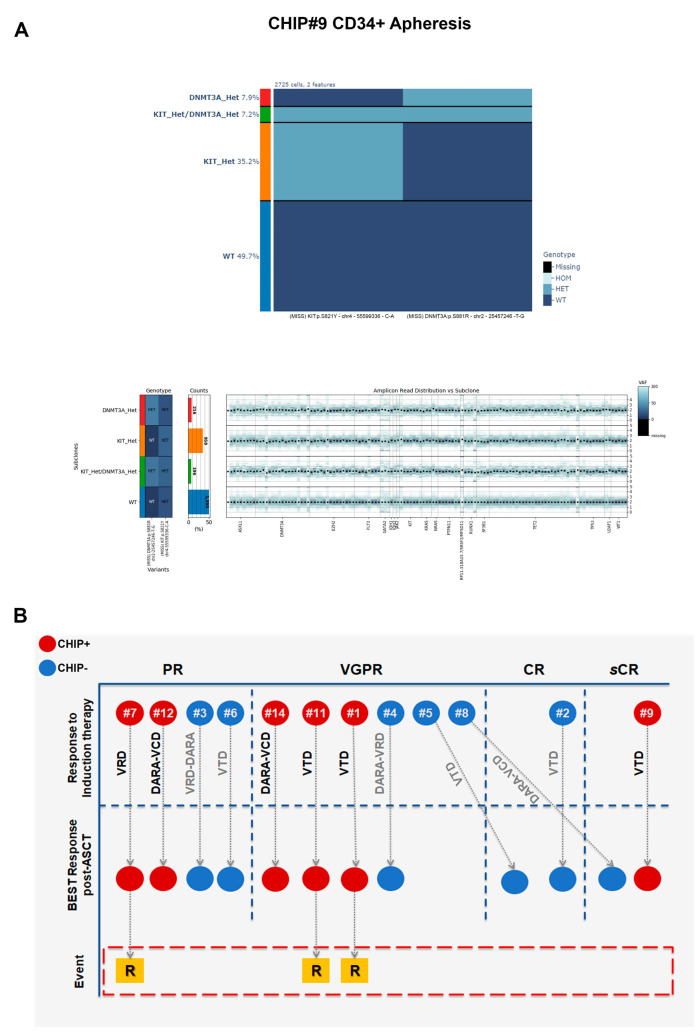
(**A**) Copy number alteration (CNA) analysis in patient CHIP#9. Variant call matrix per cell and copy number profile per clone are shown. Normalized counts of sequencing reads using the amplicons in the panel were used to calculate CNAs for each amplicon locus tested. Using the wild type (WT), heterozygous (HET), and homozygous (HOM) genotypes called by Tapestri Pipeline software, clones with no mutations (WT) were assumed to also present a normal diploid copy number and were thus used as a reference to compute potential CNAs present in mutated clones. (**B**) Clinical outcomes of MM patients. Diagram of clinical outcome of both CHIP+ and CHIP− MM patients. Response to therapy was evaluated according to the International Myeloma Working Group criteria (IMWG). CHIP+ patients are shown in red whereas CHIP− patients are shown in blue. sCR = Stringent Complete Response, CR = Complete Response, VGPR = Very Good Partial Response and PR = Partial Response. VTD = Bortezomib–Thalidomide–Dexamethasone, VRD = Bortezomib–Lenalidomide–Dexamethasone, DARA = Daratumumab, VCD = Bortezomib–Cyclophosphamide–Dexamethasone, LENA = Lenalidomide, IXA = Ixazomib, ASCT = Autologous Stem Cell Transplantation, R = Relapse.

**Table 1 cells-13-00657-t001:** Patient characteristics.

	Total (*n* = 12)	Presence of CHIP Mutation (*n* = 6)	Absence of CHIP Mutation (*n* = 6)
**Sex**			
Female	5 (42%)	2 (33%)	3 (50%)
Male	7 (58%)	4 (67%)	3 (50%)
**Median age at diagnosis**	62 (45–70)	59 (45–69)	64 (59–70)
>60 year old	8 (67%)	3 (50%)	5 (83%)
<60 year old	4 (33%)	3 (50%)	1 (17%)
**Median age at transplantation**	63 (46–70)	60 (46–69)	65 (60–67)
**ISS at diagnosis**			
I	8 (67%)	5 (83%)	3 (50%)
II	1 (8%)	0	1 (16.7%)
III	1 (8%)	0	0
Not available	2 (17%)	1 (17%)	2 (33.3%)
**First line induction therapy**			
IMiD- and PI-based triplet therapy	7 (58.3%)	4 (66.7%)	3 (50%)
DARA- and PI-based quadruplet therapy	2 (16.7%)	0	2 (33.3%)
	3 (25%)	2 (33.3%)	1 (16.7%)
**Response to therapy at the time of ASCT**			
sCR	1 (8.3%)	1 (17%)	0
CR	1 (8.3%)	0	1 (17%)
VGPR	6 (50%)	3 (50%)	3 (50%)
PR	4 (33.4%)	2 (33%)	2 (33%)
**Best response post—ASCT**			
sCR	1 (9.1%)	1 (17%)	1 (17%)
CR	1 (9.1%)	0	2 (33%)
VGPR	3 (27.3%)	3 (50%)	1 (17%)
PR	5 (45.4%)	2 (33%)	2 (33%)

IMiD = Immunomodulatory Drugs; PI = Proteasome Inhibitors; DARA = Daratumumab; sCR = Stringent Complete Response; CR = Complete Response; VGPR = Very Good Partial Response; PR = Partial Response.

## Data Availability

Data and materials described in this manuscript, including all relevant raw data, will be freely available upon reasonable request to any researcher wishing to use them for non-commercial purposes from the corresponding author.

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
