# Peer review of "Single-Cell DNA Sequencing Reveals an Evolutionary Pattern of CHIP in Transplant Eligible Multiple Myeloma Patients"

_cells, 2024, doi:10.3390/cells13080657_

Round 1

Reviewer 1 Report

Comments and Suggestions for Authors

Thank you for inviting me to review the manuscript entitled "Single-cell DNA Sequencing Reveals an Evolutionary Pattern of CHIP in Transplant Eligible Multiple Myeloma Patients". In the present study, the authors interestingly exhibited the CHIP pathogenic alterations in a number of Multiple Myeloma patients. They demonstrated the most frequently mutated genes as well as a positive selection of mutant high-fitness clones. Although this study addressed a novel concept, some issues should be considered:

1. Several parts of the Introduction can be attached together in order to increase the integrity. Consider reducing the number of paragraphs.

2. Rewrite this section as it is somehow confusing: “To determine the mutational spectrum of CHIP in patients with MM we performed a high throughput amplicon-based single-cell DNA sequencing on …”.

3. Please carefully check the whole manuscript as some mistakes and punctuational errors such as “In this study we aimed at investigating by” “typically composed by a single” exist.

Comments on the Quality of English Language

Please carefully check the whole manuscript as some mistakes and punctuational errors such as “In this study we aimed at investigating by” “typically composed by a single” exist.

Author Response

Response point-by-point to the reviewers

REVIEWER I

Comments and Suggestions for Authors

Thank you for inviting me to review the manuscript entitled "Single-cell DNA Sequencing Reveals an Evolutionary Pattern of CHIP in Transplant Eligible Multiple Myeloma Patients". In the present study, the authors interestingly exhibited the CHIP pathogenic alterations in a number of Multiple Myeloma patients. They demonstrated the most frequently mutated genes as well as a positive selection of mutant high-fitness clones. Although this study addressed a novel concept, some issues should be considered:

  1. Several parts of the Introduction can be attached together in order to increase the integrity. Consider reducing the number of paragraphs.

Answer to point 1. We thank the reviewer for the comment which has been very useful in modifying and improving the overall quality of the manuscript. The introduction section has been restricted accordingly as suggested by the reviewer and all changes have been highlighted in the revised manuscript (page 1, lines 38-40, page 2, lines 53-54 and 57-59).

  1. Rewrite this section as it is somehow confusing: “To determine the mutational spectrum of CHIP in patients with MM we performed a high throughput amplicon-based single-cell DNA sequencing on …”.

Answer to point 2. We sincerely thank the reviewer for their insightful comment, which has greatly contributed to refining and enhancing the overall quality of the manuscript. This section has been amended accordingly as suggested (page 4, lines 192-199, page 5, line 209, lines 214-216, page 6, lines 235-239 and line 251).

  1. Please carefully check the whole manuscript as some mistakes and punctuational errors such as “In this study we aimed at investigating by” “typically composed by a single” exist.

Answer to point 3. We apologize for the typo mistake. All has been corrected in the revised manuscript (page 2, lines 71-78, page 8, lines 316-321 and 325-326, page 9, lines 340-341).

Comments on the Quality of English Language

Please carefully check the whole manuscript as some mistakes and punctuational errors such as “In this study we aimed at investigating by” “typically composed by a single” exist.

Answer to Comments on the Quality of English Language. We apologize for the typo mistake. All has been corrected in the revised manuscript (page 2, lines 71-78, page 8, lines 316-321 and 325-326, page 9, lines 340-341).

Reviewer 2 Report

Comments and Suggestions for Authors

In this work, Dr. Borsi and colleagues investigate the presence of CHIP in n = 12 patients with MM, using scDNA sequencing. This study focuses on a very novel topic, using innovative techniques. The authors show a higher rate of CHIP than what was previously reported, attributing this to their more sensitive technique. One concern is the different types of treatment that the patients received as induction, which could potentially affect the study. Indeed, CHIP is evaluated not at diagnosis but from the mobilized graft. This aspect should be better elaborated in the discussion.  

Author Response

Response point-by-point to the reviewers

REVIEWER II

Comments and Suggestions for Authors

In this work, Dr. Borsi and colleagues investigate the presence of CHIP in n = 12 patients with MM, using scDNA sequencing. This study focuses on a very novel topic, using innovative techniques. The authors show a higher rate of CHIP than what was previously reported, attributing this to their more sensitive technique. One concern is the different types of treatment that the patients received as induction, which could potentially affect the study. Indeed, CHIP is evaluated not at diagnosis but from the mobilized graft. This aspect should be better elaborated in the discussion. 

Answer to Comments and Suggestions for Authors. We appreciate the reviewer's insightful comment. We acknowledge that the induction therapies administered to the patients varied in this study; however, we would like to emphasize that all patients underwent Bortezomib-based induction therapy. As far as we are aware, current literature does not provide data on the potential positive selection of mutated CHIP clones directly induced by induction therapy. Previous studies have consistently shown that clones present at diagnosis possess a proliferative advantage and intrinsic resistance to therapy, irrespective of the treatment given to patients. For example, Chitre et al. reported that 11% of MM patients exhibited evidence of CH, including 5% with VAF > 5%, prior to transplant leukapheresis. However, clonal tracing in a subset of these patients did not reveal any evidence of clonal evolution, regardless of the conditioning regimen employed. This suggests that these clones may have a selective advantage, potentially better able to withstand myeloablative conditions compared to their normal counterparts.

In another recent publication, Diamond et al. explored this hypothesis in different clinical contexts, describing two modes by which secondary malignancies, such as tMN, can arise. Notably, both modes involve the selection of a pre-existing clone by therapy. In the first mode, chemotherapy fosters the expansion of a mutated precursor by creating a conducive environment for clonal expansion. In the second mode, the pre-existing CH clone evades chemotherapy exposure during apheresis and is subsequently reintroduced through ASCT.

Although these points were briefly discussed in the Discussion section (page 15, lines 514-530 and lines 531-546), we regret not elaborating on them extensively. However, considering the size of our cohort, although significant for single-cell experiments, it may not be comprehensive enough to thoroughly address the issue. We believe that a more extensive discussion would be speculative and lacking in empirical support. Although, we greatly appreciate the reviewer’s suggestion, we feel that the section of the discussion where we mention potential scenarios of positive selection of mutated clones adequately conveys our message.

Reviewer 3 Report

Comments and Suggestions for Authors

In the current manuscript Borsi et al. describe CHIP mutation patterns in multiple myeloma patients. The manuscript provides potential useful data. However, some aspects can be improved

1-Can the authors provide data regarding the patients kariotype?

2-did the authors found additional mutations (eg BRAF V600E)?

3-CD34 isolation was performed after mobilization. Can the authors rule out that there was no clonal CD34 selection just because of the mobilization?

4-Was there any difference between CHIP+ and CHIP- patients regarding treatment resistance?

Minor points

Some parts of the result section is hard to follow. it might be convenient to do a table including the findings in the individual patients

Author Response

Response point-by-point to the reviewers

REVIEWER III

Comments and Suggestions for Authors

In the current manuscript Borsi et al. describe CHIP mutation patterns in multiple myeloma patients. The manuscript provides potential useful data. However, some aspects can be improved.

1-Can the authors provide data regarding the patients kariotype?

Answer to point 1. We thank the reviewer for bringing this to our attention. All data regarding the patient’s cytogenetic characteristics analyzed by FISH test, as well as Copy number Analysis analyzed by Ultra low pass whole genome sequencing (ULP-WGS) on tumor samples (i.e., bone marrow plasma cells), are now available in Supplementary Table 4 and Supplementary Figure 5, respectively (page 13, lines 452-454). We apologize for the inconvenience caused by our inability to conduct these additional analyses for two myeloma patients (CHIP#1 and CHIP#2, where the former tested positive for CHIP mutation while the latter tested negative) due to insufficient material.  This has been amended on Table and Figure legend. Overall, our analysis did not reveal any significant correlation between CHIP+ and CHIP- patients in terms of both CNAs and cytogenetic risk.

2-did the authors found additional mutations (eg BRAF V600E)?

Answer to point 2. We appreciate the reviewer's insightful comment. For the purpose of this study, to determine the mutational spectrum of CHIP in patients with Myeloma we performed a high throughput amplicon-based single-cell DNA sequencing on the Tapestri® platform, using a 127 amplicon panel covering 20 of the most frequently mutated genes in CHIP (i.e., SF3B1, SRSF2, U2AF1, TET2, IDH1/2, ASXL1, EZH2, DNMT3A, TP53, FLT3, NRAS, KRAS, RUNX1, C-KIT, JAK2, GATA2, PTPN11, WT1, and NPM1). We specifically chose this gene panel because our primary objective was to assess the presence of CHIP in myeloma patients, rather than to investigate the comprehensive mutational profile of these individuals. Consequently, any mutations outside of the genes covered by our panel would not have been detected.

3-CD34 isolation was performed after mobilization. Can the authors rule out that there was no clonal CD34 selection just because of the mobilization?

Answer to point 3. We thank the reviewer for bringing this to our attention. All patients enrolled in this study received a cyclophosphamide-based hematopoietic stem cell mobilization before ASCT. Consequently, all patients underwent the same mobilization therapy, suggesting that any potential clonal selection was not induced by the mobilization therapy.

4-Was there any difference between CHIP+ and CHIP- patients regarding treatment resistance?

Answer to point 4. We thank the reviewer for this question. The primary objective of our study was to detect the potential presence of CHIP in Multiple Myeloma patients at the time of transplantation and explore its correlation with therapy response. The identification of resistance mechanisms was beyond the scope of the study, and therefore, we did not investigate this aspect. Although we analyzed a considerable number of samples using a high-resolution methodology (i.e., single cell analysis), the sample size remains inadequate for conducting studies investigating possible mechanisms of resistance to therapy. Nevertheless, we did observe a trend indicating that patients with CHIP tended to experience suboptimal responses and early relapses compared to those without.

Minor points

Some parts of the result section is hard to follow. it might be convenient to do a table including the findings in the individual patients.

Answer to Minor points. We thank the reviewer to point this out. All data regarding CHIP mutational profile are now presented and summarized in Supplementary Figure 4 (page 9, line 351 and lines 356-357, page 10, line 384, page 12, line 408).

Round 2

Reviewer 1 Report

Comments and Suggestions for Authors

Dear Editor-in-chief,

The authors satisfactorily addressed all my concerns; therefore, the manuscript is ready to be published in its present form.